# The Mediating Roles of Anxiety, Depression, Sleepiness, Insomnia, and Sleep Quality in the Association between Problematic Social Media Use and Quality of Life among Patients with Cancer

**DOI:** 10.3390/healthcare10091745

**Published:** 2022-09-11

**Authors:** Vida Imani, Daniel Kwasi Ahorsu, Nasrin Taghizadeh, Zahra Parsapour, Babak Nejati, Hsin-Pao Chen, Amir H. Pakpour

**Affiliations:** 1Pediatric Health Research Center, Tabriz University of Medical Sciences, Tabriz 5166/15731, Iran; 2Sleep Disorders Unit, Department of Neurology, Acibadem University, Istanbul 34752, Turkey; 3Department of Rehabilitation Sciences, Faculty of Health & Social Sciences, The Hong Kong Polytechnic University, Hung Hom, Hong Kong; 4Medical Eye Care, Applied Sciences and Medical University, 20457 Hamburg, Germany; 5Child Growth and Development Research Center, Research Institute for Primordial Prevention of Non-Communicable Disease, Isfahan University of Medical Sciences, Isfahan 8174/673461, Iran; 6Hematology and Medical Oncology Research Center, Tabriz University of Medical Sciences, Tabriz 5166/15731, Iran; 7Division of Colon and Rectal Surgery, Department of Surgery, E-Da Hospital, Kaohsiung 824, Taiwan; 8School of Medicine, College of Medicine, I-Shou University, Kaohsiung 824, Taiwan; 9Social Determinants of Health Research Center, Research Institute for Prevention of Non-Communicable Diseases, Qazvin University of Medical Sciences, Qazvin 3419/759811, Iran; 10Department of Nursing, School of Health and Welfare, Jönköping University, SE-551 11 Jönköping, Sweden

**Keywords:** anxiety, depression, sleepiness, insomnia, sleep quality, problematic social media use, quality of life, mediation, cancer

## Abstract

The present study examined the mediating role of anxiety, depression, sleepiness, insomnia, and sleep quality in the association between problematic social media use and quality of life (QoL) among patients with cancer. This cross-sectional survey study recruited 288 patients with cancer to respond to measures on anxiety, depression, sleepiness, insomnia, sleep quality, problematic social media use, and QoL. Structural Equation Modeling was used for the mediation analysis. There were significant relationships between all of the variables used in the study. It was revealed that problematic social media use did not directly influence the QoL of patients with cancer except via anxiety, depression, sleepiness, and insomnia. Sleep quality did not mediate the association between problematic social media use and QoL. Healthcare workers managing cancer should pay attention to the mental health needs of their patients even as they treat their cancer so as to improve their quality of life. Future studies may examine other variables that affect the QoL of patients with cancer as well as other mediating and moderating variables.

## 1. Introduction

Cancer is one of the most devastating non-communicable diseases (NCDs) which has diverse effects on individuals and their immediate family members and friends. Cancer is arguably the leading cause of death with about 10 million deaths and 19.3 million new cases worldwide in 2020 [1,2,3]. The high fatality rate combined with treatment complications and socio-economic burden may leave the patient with wide-ranging psychosocial challenges [1,2,3]. Hence, mental health problems are common in cancer [4,5]. For instance, the cancer diagnosis itself may leave the patient distressed, especially when the diagnosis is poorly presented to the patient and without proper counselling [2,4,5]. This may substantially affect their mental health, quality of life, and wellbeing [2,4], even for those patients without a history of mental illness [6]. 

Evidence is well established that all patients with cancer are at risk of developing mental illness [4,6,7]. For instance, patients diagnosed with cancer have been found to have one or two mental illnesses such as depression and anxiety [4,8,9,10]. Pitman, et al. [11] reported that patients with cancer suffer from depression (20%) and anxiety (10%) compared to the general population (5% and 7% for depression and anxiety, respectively). Another study revealed that 23.4% of patients with cancer had depression, 17.7% had anxiety, 13.5% had hostility, and 9.3% had post-traumatic stress disorder [12]. Furthermore, there is a strong association between these mental illnesses generally [13,14,15,16,17]. Among patients with cancer, there have been significant associations between mental health variables such as anxiety, depression, sleepiness, insomnia, and sleep quality [12,18,19,20]. These mental illnesses further complicate treatment strategies and consequently, the effectiveness of managing cancer. Therefore, clinicians should adopt a system that monitors the mental health conditions of patients so as to benefit from prompt intervention which will further improve their well-being and quality of life (QoL). 

The well-being and QoL of patients with cancer is a very important aspect and essence of the entire treatment. The thought of regaining one’s health gives hope of enduring through the diagnosis and treatment. Cancer (including diagnosis and treatment) affects the QoL of patients negatively [21,22]. Moreover, mental illness has its fair negative effect on QoL of patients [23,24] as QoL (as used in this study) reflects the effect of an illness on an individual’s everyday life. Furthermore, there have been significant associations between QoL and mental illnesses such as anxiety [12,16,20,25], depression [12,16,20,25,26], sleepiness, insomnia [16,26], and sleep quality [20,27]. This indicates that illnesses have a higher likelihood of affecting the QoL. Therefore, in as much as medical doctors do their best to improve the physical health of patients, other health officers such as social workers and psychologists may help enhance the psychosocial aspects so as to enhance the QoL of patients with cancer. 

The current coronavirus 2019 (COVID-19) pandemic situation is aggravating the already dire healthcare needs of patients with cancer [12] due to the increased use of social media by patients as a medium of communication and/or for information [28,29,30,31]. This is particularly so due to the preventive strategies put in place to halt the spread of COVID-19 [32,33,34]. The use of physical distancing, lockdowns, and quarantining, by nature, limits the access of patients to quality healthcare [33,34,35]. These preventive strategies to halt the spread of COVID-19 also limit social support, especially from friends and families. Hence, social media is mostly and conveniently used to mitigate this challenge [28,29,30,31]. That is, patients use social media to communicate, search for information, and entertain themselves. However, social media use has its addictive side effect [36,37,38]. It is also known that problematic social media use is associated with mental illnesses [37,39,40,41,42,43] and QoL [27]. The COVID-19 pandemic era offers a unique opportunity for researchers to examine how problematic social media use associates with mental health and QoL among patients with cancer. Furthermore, as there is no known study among patients with cancer examining the mediating role of anxiety, depression, sleepiness, insomnia, and sleep quality in the association between problematic social media use and QoL among patients with cancer, this study intends to fill up the literature gap. Specifically, this study intends to examine (i) the relationships between problematic social media use, anxiety, depression, sleepiness, insomnia, sleep quality, and QoL; and (ii) the mediating role of anxiety, depression, sleepiness, insomnia, and sleep quality in the association between problematic social media use and QoL.

## 2. Materials and Methods

### 2.1. Design, Participants and Procedure

A cross-sectional survey design was used for this study. Participants recruited in this study were patients with breast cancer who have spent at least one year after their diagnosis and with a history of surgical therapy and/or chemo-radiotherapy. The data were collected using questionnaires (i.e., measures) at the center of outpatient clinics at Kowsar hospital in Qazvin and Imam Reza in Tabriz. All patients signed the informed consent before data collection. The data collection period was between March 2020 and November 2021. During this period, COVID-19 prevention policies such as restricted physical contact were in place and strictly adhered to. This study was approved by the ethics committee of Qazvin university of medical sciences (IR.QUMS.REC.1398.082). 

### 2.2. Measures 

#### 2.2.1. Short Form Health Survey-12 Item (SF-12)

The patients’ quality of life was assessed using the SF-12 [44]. It comprises 12 items with two-component (physical and mental) scores to indicate patients’ entire quality of life. Both the physical component summary (PCS) and mental component summary (MCS) scores were used for this study. This scale has acceptable psychometric properties among Iranians [45,46,47].

#### 2.2.2. Hospital Anxiety and Depression Scale (HADS)

The patients’ anxiety and depression levels were assessed using the HADS [48]. The HADS is made up of 14 items and rated on a four-point response scale format. Hence, its total score, derived from summing each item response, ranges from 0 to 21 for the anxiety and depression subscales (each subscale has seven items). The higher the score, the higher the levels of anxiety or depression. This scale has acceptable psychometric properties among Iranians [49].

#### 2.2.3. Insomnia Severity Index (ISI)

The patients’ level of insomnia was assessed using the ISI [50,51]. The ISI is made up of seven items and rated on a five-point response scale format. Its total score, derived from summing each item response, ranges from 0 to 28. The higher the score, the higher the levels of insomnia. This scale has acceptable psychometric properties among Iranians [14,52].

#### 2.2.4. Pittsburgh Sleep Quality Index (PSQI)

The patients’ quality of sleep was assessed using the PSQI [53]. The PSQI is made up of 19 items and rated on a four-point response scale format. Its total score, derived from summing each item response, ranges from 0 to 57. The higher the score, the poorer the quality of sleep. This scale has acceptable psychometric properties among Iranians [54,55,56].

#### 2.2.5. Epworth Sleepiness Scale (ESS)

The patients’ daytime sleepiness severity was assessed using the ESS [57]. The ESS is made up of eight items and rated on a four-point response scale format. Its total score, derived from summing each item response, ranges from 0 to 24. The higher the score, the greater the daytime sleepiness. This scale has acceptable psychometric properties among Iranians [58,59].

#### 2.2.6. Bergen Social Media Addiction Scale (BSMAS)

The severity of patients’ problematic social media use was assessed using the BSMAS [60]. The BSMAS is made up of six items and rated on a five-point response scale format. Its total score, derived from summing each item response, ranges from 6 to 30. The higher the score, the greater the severity of problematic social media use. This scale has acceptable psychometric properties among Iranians [61].

### 2.3. Data Analysis

The descriptive analysis was performed using frequencies and percentages as well as means (M) and standard deviations (SD). Pearson r was used to examine the relationship between the variables of the study. Structural Equation Modeling (SEM) with 5000 bias-corrected bootstraps was used for the mediation analysis. In terms of the mediation analysis, problematic social media use (assessed using BSMAS) was the independent variable; anxiety (assessed using HADS anxiety subscale), depression (assessed using HADS depression subscale), sleepiness (assessed using ESS), insomnia (assessed using ISI), and sleep quality (assessed using PSQI) were the mediating variables; and quality of life (assessed using SF-12; with its two summary scores: physical component summary and mental component summary) was the dependent variable. The multivariate normality test was performed using the Mardia test [62]. The results of the multivariate normality test were not statistically significant for both skewness and kurtosis (*p* > 0.05), suggesting that the data were normally distributed. A univariate analysis before running SEM analysis. Moreover, intercorrelation analysis between QOL and other variables was used to find which variables could be included in the SEM model. SPSS version 25 and Amos version 24 were used for performing the data analyses.

## 3. Results

The descriptive statistics in Table 1 revealed that a total of 288 participants with an average age of 52.26 years (SD = 10.44) and 5.63 years (SD = 2.94) of education participated in this study. Most of the participants were married (n = 172; 59.7%) or were single (n = 105; 36.5%). Participants have been living with the cancer diagnosis for an average of 34.2 months (SD = 17.8) with the majority currently in stage II (n = 118; 41.0%).

The correlation matrix in Table 2 revealed that there were significant positive relationships between anxiety, depression, sleepiness, insomnia, sleep quality, and problematic social media use with the correlation coefficients ranging between 0.21 and 0.64 (*p* < 0.001). The physical component summary or mental component summary related significantly negative with the other variables with the correlation coefficients ranging between −0.20 and −0.53 (*p* < 0.001). The physical component summary related significantly positive with mental component summary (*r* = 0.53, *p* < 0.001).

The mediation analysis in Table 3 revealed that there was no direct association between problematic social media use and the two sub-components of participants’ general health: physical component summary (standardized coefficient = −0.008, *p* = 0.998) and mental component summary (standardized coefficient = −0.021, *p* = 0.728). However, there were significant total effect of problematic social media use on physical component summary (standardized coefficient = −0.723, *p* = 0.005) and mental component summary (standardized coefficient = −0.872, *p* = 0.007). It was found that anxiety (standardized coefficient = −0.132, 95% CI = −0.248, −0.060), depression (standardized coefficient = −0.100, 95% CI = −0.211, −0.045), sleepiness (standardized coefficient = −0.110, 95% CI = −0.219, −0.035), and insomnia (standardized coefficient = −0.175, 95% CI = −0.285, −0.079) mediated the association between problematic social media use and physical component summary. Moreover, anxiety (standardized coefficient = −0.172, 95% CI = −0.282, −0.083), depression (standardized coefficient = −0.110, 95% CI = −0.188, −0.037), sleepiness (standardized coefficient = −0.088, 95% CI = −0.179, −0.015), and insomnia (standardized coefficient = −0.169, 95% CI = −0.271, −0.084) mediated the association between problematic social media use and mental component summary. Figure 1 shows the association between problematic social media use, anxiety, depression, sleepiness, insomnia, sleep quality, and QoL (i.e., physical component summary and mental component summary).

## 4. Discussion

The present study, which examined the mediating roles of anxiety, depression, sleepiness, insomnia, and sleep quality in the association between problematic social media use and QoL among patients with cancer, revealed that problematic social media use only indirectly affected QoL via anxiety, depression, sleepiness and insomnia but not sleep quality.

Specifically, the results from the correlation matrix indicated that there were significant relationships between all the variables used in the study. There were three ways in which these significant relationships manifested. The first was positive relationships between anxiety, depression, sleepiness, insomnia, sleep quality, and problematic social media use. This suggests that as one of these variables increases, the other variable may also increase and vice versa. The second was negative relationships between physical component summary or mental component summary (i.e., QoL) and anxiety, depression, sleepiness, insomnia, sleep quality, and problematic social media use. This suggests that as QoL (i.e., physical component summary or mental component summary) increases, the other variables may decrease and vice versa. The third was the relationship between the two components of QoL. There was a positive relationship between physical component summary and mental component summary which suggests that as physical component summary increases, mental component summary may also increase. These findings confirm the findings of previous studies [12,18,19,20,26] which further indicate that mental health challenges influence the QoL among patients with cancer. 

The mediation results revealed that anxiety, depression, sleepiness, and insomnia mediated the association between problematic social media use and physical component summary. Moreover, anxiety, depression, sleepiness, and insomnia mediated the association between problematic social media use and mental component summary. That is, sleep quality was the only variable that did not mediate the association between problematic social media use and physical component summary or mental component summary. These results are particularly important as there were no direct association between problematic social media use and physical component summary or mental component summary. Therefore, the mediation results indicate that the only way problematic social media use associates with the QoL of patients with cancer is via anxiety, depression, sleepiness, and insomnia. Hence, healthcare officers, such as doctors, nurses, and psychologists, should work collectively to help reduce the levels of anxiety, depression, sleepiness, and insomnia among patients living with cancer. These mediation findings are unique as no known study has examined this mediation although depression, anxiety, and insomnia are known factors that do mediate other associations [25,26]. This reaffirms our earlier assertion that there should be a multidisciplinary healthcare team managing patients with cancer so as to harness the benefits of holistic cancer management. Specifically, patients with cancer may benefit from (i) psychoeducation on problematic social media use and its effect on their health and QoL from health officers such as psychologists and medical doctors, and (ii) exercise and/or relaxation therapy to improve their mental health and QoL from health officers such as physiotherapists, occupational therapists, and psychologists. 

### Limitation

This study used a cross-sectional survey design which limits the causality evidence of the association between the variables although SEM analysis may suggest so. A longitudinal study may help to ascertain the causality of these associations as well as how the variables change over time. As COVID-19 prevention protocols are applied differently in different countries, these findings should be applied with caution. Furthermore, replication is recommended to enhance the understanding and generalizability of the findings. Similarly, factors that affected prompt and quality service delivery during COVID-19 and the number of surgeries were not reported in this study. This would have helped to situate the challenges patients with cancer faced during COVID-19. Lastly, self-report measures were used which are known to have limitations such as social desirability response bias. However, the SEM analysis and the robust psychometric properties do suggest that data and findings are valid and can be trusted.

## 5. Conclusions

The present study examined the mediating role of anxiety, depression, sleepiness, insomnia, and sleep quality in the association between problematic social media use and QoL among patients with cancer. Although the findings indicated that there were significant relationships between the variables, the mediation analysis revealed that problematic social media use did not directly influence the QoL of patients with cancer except via anxiety, depression, sleepiness, and insomnia. This implies that healthcare workers should pay attention to how patients depend on electronic devices, the internet, and social media in order to properly educate them on the potential effect of its addiction on their mental health and QoL. Future studies may examine other variables that affect the QoL of patients with cancer as well as other mediating and moderating variables. 

## Figures and Tables

**Figure 1 healthcare-10-01745-f001:**
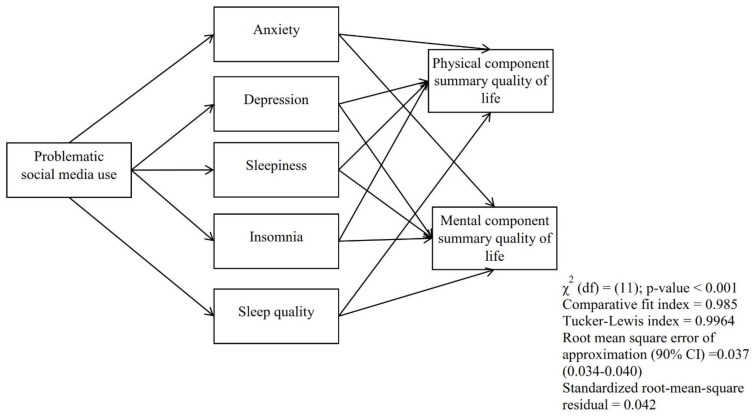
The mediation model via structural equation modeling (SEM) showing the mediating roles of anxiety, depression, sleepiness, insomnia, and sleep quality in the association between problematic social media use and quality of life among patients with cancer.

**Table 1 healthcare-10-01745-t001:** Characteristics of the participants (N = 288).

	Mean (SD) or n (%)
Age (in years)	52.26 (10.44)
Education (in years)	5.63 (2.94)
Marital status	
Single	105 (36.5%)
Married	172 (59.7%)
Widowed/divorced	11 (3.8%)
Time since diagnosis (in months)	34.2 (17.8)
Stage at diagnosis	
0	45 (15.6%)
I	63 (21.9%)
II	118 (41.0%)
III	62 (21.5%)

**Table 2 healthcare-10-01745-t002:** Correlation matrix between the study’s variables.

Study Variable	1	2	3	4	5	6	7	8	Mean	SD
Anxiety	1	0.64	0.40	0.48	0.24	0.34	−0.41	−0.53	10.46	4.93
Depression		1	0.26	0.41	0.31	0.31	−0.35	−0.40	11.81	6.79
Sleepiness			1	0.54	0.23	0.41	−0.31	−0.29	10.07	4.35
Insomnia				1	0.43	0.40	−0.45	−0.46	7.83	5.34
Sleep quality					1	0.21	−0.26	−0.29	6.35	4.72
Problematic Social media use						1	−0.20	−0.26	16.72	7.57
Physical component summary							1	0.53	45.61	27.32
Mental component summary								1	50.55	25.29

All *p*-values < 0.001.

**Table 3 healthcare-10-01745-t003:** Standardized effects (including direct, indirect, and total effects) of the relationship between problematic social media use and health-related quality of life.

Parameter	Total Effect (*p*-Value)	Standardized Direct Effect (*p*-Value)	Standardized Indirect Effect (95% CI)	Bootstrapping SE (*p*-Value) ^a^
BSMAS → ISI→PCS	-	−0.028 (0.666)	−0.175 (−0.285, −0.079)	0.049 (0.004)
BSMAS →ESS→PCS	-	−0.093 (0.171)	−0.110 (−0.219, −0.035)	0.048 (0.012)
BSMAS → PSQI→PCS	-	−0.155 (0.090)	−0.048 (−0.197,0.001)	0.062 (0.068)
BSMAS → Anxiety →PCS	-	−0.071 (0.270)	−0.132 (−0.248, −0.060)	0.046 (0.002)
BSMAS → Depression →PCS	-	−0.103 (0.148)	−0.100 (−0.211, −0.045)	0.036 (0.002)
BSMAS → PCS	−0.732 (0.005)	−0.008 (0.998)	−0.198 (−0.319, −0.083)	0.058 (0.005)
BSMAS → ISI→MCS	-	−0.092 (0.172)	−0.169 (−0.271, −0.084)	0.048 (0.003)
BSMAS →ESS→MCS	-	−0.173 (0.036)	−0.088 (−0.179, −0.015)	0.048 (0.020)
BSMAS → PSQI→MCS	-	−0.211 (0.039)	−0.051 (−0.200,00.1)	0.062 (0.093)
BSMAS → Anxiety →MCS	-	−0.090 (0.123)	−0.172 (−0.282, −0.083)	0.051 (0.003)
BSMAS → Depression →MCS	-	−0.152 (0.050)	−0.110 (−0.188, −0.037)	0.039 (0.006)
BSMAS → MCS	−0.872 (0.007)	−0.021 (0.728)	−0.253 (−0.381, −0.118)	0.065 (0.007)

Note: Sociodemographic and clinical variables in Table 1 were adjusted for in the model. ^a^ Bootstrapping SE and its *p*-value reflect the standard error and significance for only indirect effect. BSMAS = Bergen Social Media Addiction Scale. ISI = Insomnia Severity Index. ESS = Epworth Sleepiness Scale. PSQI = Pittsburgh Sleep Quality Index. PCS = Physical Component Summary. MCS = Mental Component Summary. Confidence intervals were calculated using a bootstrap resampling method (n = 5000).

## Data Availability

The data are not publicly available due to data restriction policies.

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
