# Peer review of "The Mediating Roles of Anxiety, Depression, Sleepiness, Insomnia, and Sleep Quality in the Association between Problematic Social Media Use and Quality of Life among Patients with Cancer"

_healthcare, 2022, doi:10.3390/healthcare10091745_

Round 1

Reviewer 1 Report

Revision of the article:

The mediating roles of anxiety, depression, sleepiness, insomnia, and sleep quality in the association between problematic social media use and quality of life among patients with cancer

This article aims to examine the mediating role of anxiety, depression, sleepiness, insomnia, and sleep quality in the association between problematic social media use and Quality of Life among patients with cancer.

This is a well-designed and well-developed study, although, according to what the authors reflect in the limitations, having carried out a longitudinal study would have allowed them to establish greater evidence of causality of the association between the variables.

Although it is a good article, prior to its acceptance for publication, it would be important to correct and improve the following aspects:

·        The references section should be reviewed, since it does not follow the recommendations of the journal, such as writing the name of the journal with the abbreviation (for example in reference number 8, 10, 14, 16, 17, 23, 26, 27, etc). In addition, in some references the volume of the journal is missing, as is the case of reference number 30. Please review the recommendations of the journal at the following link: https://www.mdpi.com/journal/healthcare/instructions

·        In order to take care of the presentation, it is requested to reduce the font size of table 2, so that it is homogeneous to that used in table 1 and table 3.

·        In the results section, it would be important to include a reference to tables 1, 2 and 3 in the text.

·        Taking into account the results and what are the mediating variables in the association between problematic social media use and Quality of Life among cancer patients, a greater specification of the implications for practice is lacking. It would be desirable to include some proposal for interdisciplinary interventions to help people with cancer to improve their quality of life.

·        Please complete the acknowledgments section or delete it.

I hope these suggestions will help you to improve the article.

Kind regards

Author Response

Itemised reply to Reviewer 1

The mediating roles of anxiety, depression, sleepiness, insomnia, and sleep quality in the association between problematic social media use and quality of life among patients with cancer

This article aims to examine the mediating role of anxiety, depression, sleepiness, insomnia, and sleep quality in the association between problematic social media use and Quality of Life among patients with cancer.

This is a well-designed and well-developed study, although, according to what the authors reflect in the limitations, having carried out a longitudinal study would have allowed them to establish greater evidence of causality of the association between the variables.

Although it is a good article, prior to its acceptance for publication, it would be important to correct and improve the following aspects:

  1. The references section should be reviewed, since it does not follow the recommendations of the journal, such as writing the name of the journal with the abbreviation (for example in reference number 8, 10, 14, 16, 17, 23, 26, 27, etc). In addition, in some references the volume of the journal is missing, as is the case of reference number 30. Please review the recommendations of the journal at the following link: https://www.mdpi.com/journal/healthcare/instructions

Reply: We thank the reviewer for the candid assessment and comment. We have revised the reference section of the manuscript.

Lines 315-490: “References

  1. World Health Organisation. Cancer. Available online: https://www.who.int/news-room/fact-sheets/detail/cancer (accessed on 24th July 2022).
  2. Sung, H.; Ferlay, J.; Siegel, R.L.; Laversanne, M.; Soerjomataram, I.; Jemal, A.; Bray, F. Global cancer statistics 2020: GLOBOCAN estimates of incidence and mortality worldwide for 36 cancers in 185 countries. CACancer J Clin 2021, 71, 209-249, https://doi.org/10.3322/caac.21660.
  3. Bray, F.; Laversanne, M.; Weiderpass, E.; Soerjomataram, I. The ever-increasing importance of cancer as a leading cause of premature death worldwide. Cancer 2021, 127, 3029-3030, doi:10.1002/cncr.33587.
  4. Niedzwiedz, C.L.; Knifton, L.; Robb, K.A.; Katikireddi, S.V.; Smith, D.J. Depression and anxiety among people living with and beyond cancer: A growing clinical and research priority. BMC Cancer 2019, 19, 943, doi:10.1186/s12885-019-6181-4.
  5. Ugalde, A.; Haynes, K.; Boltong, A.; White, V.; Krishnasamy, M.; Schofield, P.; Aranda, S.; Livingston, P. Self-guided interventions for managing psychological distress in people with cancer – A systematic review. Patient Educ Couns 2017, 100, 846-857, https://doi.org/10.1016/j.pec.2016.12.009.
  6. Zhu, J.; Fang, F.; Sjölander, A.; Fall, K.; Adami, H.O.; Valdimarsdóttir, U. First-onset mental disorders after cancer diagnosis and cancer-specific mortality: A nationwide cohort study. Ann Oncol 2017, 28, 1964-1969, doi:10.1093/annonc/mdx265.
  7. Wikman, A.; Ljung, R.; Johar, A.; Hellstadius, Y.; Lagergren, J.; Lagergren, P. Psychiatric morbidity and survival after surgery for esophageal cancer: A population-based cohort study. J Clin Oncol 2015, 33, 448-454, doi:10.1200/jco.2014.57.1893.
  8. Mitchell, A.J.; Chan, M.; Bhatti, H.; Halton, M.; Grassi, L.; Johansen, C.; Meader, N. Prevalence of depression, anxiety, and adjustment disorder in oncological, haematological, and palliative-care settings: A meta-analysis of 94 interview-based studies. Lancet Oncol 2011, 12, 160-174, https://doi.org/10.1016/S1470-2045(11)70002-X.
  9. Nakash, O.; Levav, I.; Aguilar-Gaxiola, S.; Alonso, J.; Andrade, L.H.; Angermeyer, M.C.; Bruffaerts, R.; Caldas-de-Almeida, J.M.; Florescu, S.; de Girolamo, G.; et al. Comorbidity of common mental disorders with cancer and their treatment gap: Findings from the World Mental Health Surveys. Psychooncology 2014, 23, 40-51, doi:10.1002/pon.3372.
  10. Naser, A.Y.; Hameed, A.N.; Mustafa, N.; Alwafi, H.; Dahmash, E.Z.; Alyami, H.S.; Khalil, H. Depression and anxiety in patients with cancer: A cross-sectional study. Front Psychol 2021, 12, doi:10.3389/fpsyg.2021.585534.
  11. Pitman, A.; Suleman, S.; Hyde, N.; Hodgkiss, A. Depression and anxiety in patients with cancer. BMJ 2018, 361, k1415, doi:10.1136/bmj.k1415.
  12. Wang, Y.; Duan, Z.; Ma, Z.; Mao, Y.; Li, X.; Wilson, A.; Qin, H.; Ou, J.; Peng, K.; Zhou, F.; et al. Epidemiology of mental health problems among patients with cancer during COVID-19 pandemic. Transl Psychiatry 2020, 10, 263, doi:10.1038/s41398-020-00950-y.
  13. Ahorsu, D.K.; Lin, C.-Y.; Alimoradi, Z.; Griffiths, M.D.; Chen, H.-P.; Broström, A.; Timpka, T.; Pakpour, A.H. Cyberchondria, fear of covid-19, and risk perception mediate the association between problematic social media use and intention to get a COVID-19 vaccine. Vaccines 2022, 10, 122, doi:10.3390/vaccines10010122.
  14. Ahorsu, D.K.; Lin, C.-Y.; Pakpour, A.H. The association between health status and insomnia, mental health, and preventive behaviors: The mediating role of fear of COVID-19. Gerontol Geriatr Med 2020, 6, 2333721420966081, doi:10.1177/2333721420966081.
  15. Ahorsu, D.K.; Pramukti, I.; Strong, C.; Wang, H.-W.; Griffiths, M.D.; Lin, C.-Y.; Ko, N.-Y. COVID-19-related variables and its association with anxiety and suicidal ideation: Differences between international and local university students in Taiwan. Psychol Res Behav Manag 2021, 14, 1857-1866, doi:10.2147/PRBM.S333226.
  16. Fazeli, S.; Mohammadi Zeidi, I.; Lin, C.-Y.; Namdar, P.; Griffiths, M.D.; Ahorsu, D.K.; Pakpour, A.H. Depression, anxiety, and stress mediate the associations between internet gaming disorder, insomnia, and quality of life during the COVID-19 outbreak. Addict Behav Rep 2020, 12, 100307, https://doi.org/10.1016/j.abrep.2020.100307.
  17. Lu, M.-Y.; Ahorsu, D.K.; Kukreti, S.; Strong, C.; Lin, Y.-H.; Kuo, Y.-J.; Chen, Y.-P.; Lin, C.-Y.; Chen, P.-L.; Ko, N.-Y.; et al. The prevalence of post-traumatic stress disorder symptoms, sleep problems, and psychological distress among COVID-19 frontline healthcare workers in Taiwan. Front Psychiatry 2021, 12, 705657-705657, doi:10.3389/fpsyt.2021.705657.
  18. Cho, O.H.; Hwang, K.H. Association between sleep quality, anxiety and depression among Korean breast cancer survivors. Nurs Open 2021, 8, 1030-1037, doi:10.1002/nop2.710.
  19. Ho, S.Y.; Rohan, K.J.; Parent, J.; Tager, F.A.; McKinley, P.S. A longitudinal study of depression, fatigue, and sleep disturbances as a symptom cluster in women with breast cancer. J Pain Symptom Manage 2015, 49, 707-715, doi:10.1016/j.jpainsymman.2014.09.009.
  20. Emre, N.; Yılmaz, S. Sleep quality, mental health, and quality of life in women with breast cancer. Indian J Cancer 2022, doi:10.4103/ijc.IJC_859_20.
  21. Nayak, M.G.; George, A.; Vidyasagar, M.S.; Mathew, S.; Nayak, S.; Nayak, B.S.; Shashidhara, Y.N.; Kamath, A. Quality of life among cancer patients. Indian J Palliat Care 2017, 23, 445-450, doi:10.4103/ijpc.Ijpc_82_17.
  22. Ramasubbu, S.K.; Pasricha, R.K.; Nath, U.K.; Rawat, V.S.; Das, B. Quality of life and factors affecting it in adult cancer patients undergoing cancer chemotherapy in a tertiary care hospital. Cancer Rep 2021, 4, e1312, https://doi.org/10.1002/cnr2.1312.
  23. Berghöfer, A.; Martin, L.; Hense, S.; Weinmann, S.; Roll, S. Quality of life in patients with severe mental illness: A cross-sectional survey in an integrated outpatient health care model. Qual Life Res 2020, 29, 2073-2087, doi:10.1007/s11136-020-02470-0.
  24. Pascual-Sánchez, A.; Jenaro, C.; Montes-Rodríguez, J.M. Quality of life in euthymic bipolar patients: A systematic review and meta-analysis. J Affect Disord 2019, 255, 105-115, doi:10.1016/j.jad.2019.05.032.
  25. Charalambous, A.; Giannakopoulou, M.; Bozas, E.; Paikousis, L. Parallel and serial mediation analysis between pain, anxiety, depression, fatigue and nausea, vomiting and retching within a randomised controlled trial in patients with breast and prostate cancer. BMJ Open 2019, 9, e026809, doi:10.1136/bmjopen-2018-026809.
  26. Cha, K.M.; Chung, Y.K.; Lim, K.Y.; Noh, J.S.; Chun, M.; Hyun, S.Y.; Kang, D.R.; Oh, M.J.; Kim, N.H. Depression and insomnia as mediators of the relationship between distress and quality of life in cancer patients. J Affect Disord 2017, 217, 260-265, https://doi.org/10.1016/j.jad.2017.04.020.
  27. Karimy, M.; Parvizi, F.; Rouhani, M.R.; Griffiths, M.D.; Armoon, B.; Fattah Moghaddam, L. The association between internet addiction, sleep quality, and health-related quality of life among Iranian medical students. J Addict Dis 2020, 38, 317-325, doi:10.1080/10550887.2020.1762826.
  28. Gudiño, D.; Fernández-Sánchez, M.J.; Becerra-Traver, M.T.; Sánchez, S. Social media and the pandemic: Consumption habits of the Spanish population before and during the COVID-19 lockdown. Sustainability 2022, 14, doi:10.3390/su14095490.
  29. Pandya, A.; Lodha, P. Social connectedness, excessive screen time during COVID-19 and mental health: A review of current evidence. Front Hum Dyn 2021, 3, doi:10.3389/fhumd.2021.684137.
  30. Pérez-Escoda, A.; Jiménez-Narros, C.; Perlado-Lamo-de-Espinosa, M.; Pedrero-Esteban, L.M. Social networks' engagement during the COVID-19 pandemic in Spain: Health media vs. healthcare professionals. Int J Environ Res Public Health 2020, 17, doi:10.3390/ijerph17145261.
  31. Patel, V.; Chaudhary, P.; Kumar, P.; Vasavada, D.; Tiwari, D. A study of correlates of social networking site addiction among the undergraduate health professionals. Asian J Soc Health Behav 2021, 4, 30-35, doi:10.4103/shb.shb_1_21.
  32. Lin, M.-W.; Cheng, Y. Policy actions to alleviate psychosocial impacts of COVID-19 pandemic: Experiences from Taiwan. Soc Health Behav 2020, 3, 72-73, doi:10.4103/shb.Shb_18_20.
  33. Güner, R.; HasanoÄŸlu, I.; AktaÅŸ, F. COVID-19: Prevention and control measures in community. Turk J Med Sci 2020, 50, 571-577, doi:10.3906/sag-2004-146.
  34. Gyasi, R.M. Fighting COVID-19: Fear and internal conflict among older adults in Ghana. J Gerontol Soc Work 2020, 63, 688-690, doi:10.1080/01634372.2020.1766630.
  35. Hao, F.; Tan, W.; Jiang, L.; Zhang, L.; Zhao, X.; Zou, Y.; Hu, Y.; Luo, X.; Jiang, X.; McIntyre, R.S.; et al. Do psychiatric patients experience more psychiatric symptoms during COVID-19 pandemic and lockdown? A case-control study with service and research implications for immunopsychiatry. Brain Behav Immun 2020, 87, 100-106, doi:10.1016/j.bbi.2020.04.069.
  36. Twenge, J.M.; Campbell, W.K. Media use is linked to lower psychological well-being: Evidence from three datasets. Psychiatr Q 2019, 90, 311-331, doi:10.1007/s11126-019-09630-7.
  37. Chen, I.-H.; Ahorsu, D.K.; Pakpour, A.H.; Griffiths, M.D.; Lin, C.-Y.; Chen, C.-Y. Psychometric properties of three simplified Chinese online-related addictive behavior instruments among mainland Chinese primary school students. Front Psychiatry 2020, 11, doi:10.3389/fpsyt.2020.00875.
  38. Ranjan, L.; Gupta, P.; Srivastava, M.; Gujar, N. Problematic internet use and its association with anxiety among undergraduate students. Asian J Soc Health Behav 2021, 4, 137-141, doi:10.4103/shb.shb_30_21.
  39. Zhao, J.; Jia, T.; Wang, X.; Xiao, Y.; Wu, X. Risk factors associated with social media addiction: An exploratory study. Front Psychol 2022, 13, 837766, doi:10.3389/fpsyg.2022.837766.
  40. Shannon, H.; Bush, K.; Villeneuve, P.J.; Hellemans, K.G.C.; Guimond, S. Problematic social media use in adolescents and young adults: Systematic review and meta-analysis. JMIR Ment Health 2022, 9, e33450, doi:10.2196/33450.
  41. Lin, C.-Y.; Potenza, M.N.; Ulander, M.; Broström, A.; Ohayon, M.M.; Chattu, V.K.; Pakpour, A.H. Longitudinal relationships between nomophobia, addictive use of social media, and insomnia in adolescents. Healthcare (Basel) 2021, 9, 1201, doi:10.3390/healthcare9091201.
  42. Abiddine, F.Z.E.; Aljaberi, M.A.; Gadelrab, H.F.; Lin, C.-Y.; Muhammed, A. Mediated effects of insomnia in the association between problematic social media use and subjective well-being among university students during COVID-19 pandemic. Sleep Epidemiol 2022, 2, 100030, https://doi.org/10.1016/j.sleepe.2022.100030.
  43. Wong, H.Y.; Mo, H.Y.; Potenza, M.N.; Chan, M.N.M.; Lau, W.M.; Chui, T.K.; Pakpour, A.H.; Lin, C.-Y. Relationships between severity of internet gaming disorder, severity of problematic social media use, sleep quality and psychological distress. Int J Environ Res Public Health 2020, 17, 1879, doi:10.3390/ijerph17061879.
  44. Ware Jr, J.E.; Kosinski, M.; Keller, S.D. A 12-Item Short-Form Health Survey: Construction of scales and preliminary tests of reliability and validity. Med Care 1996, 34, 220-233.
  45. Montazeri, A.; Vahdaninia, M.; Mousavi, S.J.; Asadi-Lari, M.; Omidvari, S.; Tavousi, M. The 12-item medical outcomes study short form health survey version 2.0 (SF-12v2): A population-based validation study from Tehran, Iran. Health Qual Life Outcomes 2011, 9, 12-12, doi:10.1186/1477-7525-9-12.
  46. Pakpour, A.H.; Nourozi, S.; Molsted, S.; Harrison, A.P.; Nourozi, K.; Fridlund, B. Validity and reliability of short form-12 questionnaire in Iranian hemodialysis patients. Iran J Kidney Dis 2011, 5, 175-181.
  47. Ahorsu, D.K.; Lin, C.-Y.; Marznaki, Z.H.; H. Pakpour, A. The association between fear of COVID-19 and mental health: The mediating roles of burnout and job stress among emergency nursing staff. Nurs Open 2022, 9, 1147-1154, doi:https://doi.org/10.1002/nop2.1154.
  48. Zigmond, A.S.; Snaith, R.P. The hospital anxiety and depression scale. Acta Psychiatr Scand 1983, 67, 361-370.
  49. Montazeri, A.; Vahdaninia, M.; Ebrahimi, M.; Jarvandi, S. The hospital anxiety and depression scale (HADS): Translation and validation study of the Iranian version. Health Qual Life Outcomes 2003, 1, 14, doi:10.1186/1477-7525-1-14.
  50. Bastien, C.H.; Vallières, A.; Morin, C.M. Validation of the insomnia severity index as an outcome measure for insomnia research. Sleep Med 2001, 2, 297-307, https://doi.org/10.1016/S1389-9457(00)00065-4.
  51. Morin, C.M. Insomnia: Psychological assessment and management; Guilford press: NY, USA,1993.
  52. Yazdi, Z.; Sadeghniiat-Haghighi, K.; Zohal, M.A.; Elmizadeh, K. Validity and reliability of the Iranian version of the insomnia severity index. Malays J Med Sci 2012, 19, 31-36.
  53. Buysse, D.J.; Reynolds, C.F.; Monk, T.H.; Berman, S.R.; Kupfer, D.J. The Pittsburgh sleep quality index: A new instrument for psychiatric practice and research. Psychiatry Res 1989, 28, 193-213, doi:https://doi.org/10.1016/0165-1781(89)90047-4.
  54. Majd, N.R.; Broström, A.; Ulander, M.; Lin, C.-Y.; Griffiths, M.D.; Imani, V.; Ahorsu, D.K.; Ohayon, M.M.; Pakpour, A.H. Efficacy of a theory-based cognitive behavioral technique app-intervention for patients with insomnia: A randomized controlled trial. J Med Internet Res 2020, 22(4), e15841, doi:10.2196/15841.
  55. Farrahi Moghaddam, J.; Nakhaee, N.; Sheibani, V.; Garrusi, B.; Amirkafi, A. Reliability and validity of the Persian version of the Pittsburgh sleep quality index (PSQI-P). Sleep Breath 2012, 16, 79-82, doi:10.1007/s11325-010-0478-5.
  56. Carpenter, J.S.; Andrykowski, M.A. Psychometric evaluation of the Pittsburgh sleep quality index. J Psychosom Res 1998, 45, 5-13, doi:10.1016/s0022-3999(97)00298-5.
  57. Johns, M.W. A new method for measuring daytime sleepiness: The Epworth sleepiness scale. Sleep 1991, 14, 540-545, doi:10.1093/sleep/14.6.540.
  58. Sadeghniiat Haghighi, K.; Montazeri, A.; Khajeh Mehrizi, A.; Aminian, O.; Rahimi Golkhandan, A.; Saraei, M.; Sedaghat, M. The Epworth Sleepiness Scale: Translation and validation study of the Iranian version. Sleep Breath 2013, 17, 419-426, doi:10.1007/s11325-012-0646-x.
  59. Lin, C.-Y.; Imani, V.; Griffiths, M.D.; Broström, A.; Nygårdh, A.; Demetrovics, Z.; Pakpour, A.H. Temporal associations between morningness/eveningness, problematic social media use, psychological distress and daytime sleepiness: Mediated roles of sleep quality and insomnia among young adults. J Sleep Res 2021, 30, e13076, https://doi.org/10.1111/jsr.13076.
  60. Andreassen, C.S.; Billieux, J.; Griffiths, M.D.; Kuss, D.J.; Demetrovics, Z.; Mazzoni, E.; Pallesen, S. The relationship between addictive use of social media and video games and symptoms of psychiatric disorders: A large-scale cross-sectional study. Psychol Addict Behav 2016, 30, 252-262, doi:10.1037/adb0000160.
  61. Lin, C.-Y.; Broström, A.; Nilsen, P.; Griffiths, M.; Pakpour, A. Psychometric validation of the Persian Bergen social media addiction scale using classic test theory and Rasch models. J Behav Addict 2017, 6, 620-629.
  62. Mardia, K.V. Measures of multivariate skewness and kurtosis with applications. Biometrika 1970, 57(3), 519-530.”

  1. In order to take care of the presentation, it is requested to reduce the font size of table 2, so that it is homogeneous to that used in table 1 and table 3.

Reply: We thank the reviewer for the suggestion. We have reduced the font size of Table 2. Please see Page 5 of the manuscript.

  1. In the results section, it would be important to include a reference to tables 1, 2 and 3 in the text.

Reply: We thank the reviewer for the kind comment. We have revised the result section by stating the Table number at where their results are being reported.

Lines 171-173: “The descriptive statistics in Table 1 revealed that a total of 288 participants with an average age of 52.26 years (SD = 10.44) and 5.63 years (SD = 2.94) of education participated in this study”

Lines 178-181: “The correlation matrix in Table 2 revealed that there were significant positive relationships between anxiety, depression, sleepiness, insomnia, sleep quality, and problematic social media use with the correlation coefficients ranging between 0.21 and 0.64 (ps < 0.001).”

Lines 188-191: “The mediation analysis in Table 3 revealed that there was no direct association between problematic social media use and the two sub-components of participants’ general health; physical component summary (standardized coefficient = -0.008, p = 0.998) and mental component summary (standardized coefficient = -0.021, p = 0.728).”

  1. Taking into account the results and what are the mediating variables in the association between problematic social media use and Quality of Life among cancer patients, a greater specification of the implications for practice is lacking. It would be desirable to include some proposal for interdisciplinary interventions to help people with cancer to improve their quality of life.

Reply: We thank the reviewer for the suggestion. We have specifically added a proposal on how interdisciplinary interventions will help improve the mental health and QoL of people with cancer.

Lines 262-268: “This reaffirms our earlier assertion that there should be a multidisciplinary healthcare team managing patients with cancer so as to harness the benefits of holistic cancer management. Specifically, patients with cancer may benefit from (i) psychoeducation on problematic social media use and its effect on their health and QoL from health officers such as psychologists and medical doctors, and (ii) exercise and/or relaxation therapy to improve their mental health and QoL from health officers such as physiotherapists, occupational therapists, and psychologists.”

  1. Please complete the acknowledgments section or delete it.

Reply: We have deleted the acknowledgment section from the manuscript.

I hope these suggestions will help you to improve the article.

Kind regards

Reply: We are very grateful for your insightful and constructive comments. I really did help.

Reviewer 2 Report

This article provides strong evidence for relationships between social media use, mental health and sleep measures, and quality of life in a population of breast cancer patients under post-surgery surveillance.  The topic is an important one, and it addresses important patient factors for recovery in a significant patient population.  The sample size is sufficient, the measures are appropriate, and the analysis appears to have been carried out correctly.

I do think that the authors could make the article much more approachable and impactful, by better describing the context and background and by better displaying and describing the results.  As written, the article feels like a pretty run-of-the-mill analysis of typical survey measures; but considering the unique population studied, as well as the novelty of studying social media use, I don’t think that this needs to be the case.  Please see below for some more detailed suggestions.

·         ll. 46-51.  The end of this first par is a bit convoluted.  Better to start with a simple statement (and refs) that mental health problems are more common in cancer.  (As in the next par.)  Then move on to why that might be the case and what the consequences are.

·         L. 52.  “most patients with cancer are at risk”   Are all patients at risk?  Do most patients develop mental illness?  I’m not sure I see why most, but not all, would be at risk.

·         Ll. 63-65.  The claim that holistic approaches should be adopted is not really supported.  I think you could support that systems should monitor MH and social media use; but it’s not clear that that entails ‘holistic.’

·         Ll 66-76.  I think it would help to make a clearer distinction between the concepts of MH and QoL, and to explain the relationship, so that this doesn’t feel like repetition.  Certainly some QoL items are, basically, about MH. 

·         Ll. 86-88.  If introducing covid, then it seems important to more directly make a case for either increased use of social media by patients during the pandemic and/or increased impacts of use.  The introduction of covid here feels a bit unfocussed.

·         Ll 89-95.  Seems like you should provide some references for effects of social media on these various outcomes.  I’m sure they’re out there; but you haven’t particularly covered that angle.

·         It’ not clear how the timing of data collection and disease even overlaps with the covid-19 pandemic.  It appears that many of the participants would have been in the hospital before covid.  And data collection is long after hospitalization, when some patients are probably recovering nicely.  If covid really is a factor, then I think the authors should try to address that timing issue, as well as the heterogeneity of their sample in that respect.

·         2.3:  It’s advisable to at least consider variable distributions, and to think about whether some non-parametric tests or transformations might be called for.

·         2.3:  A diagram depicting the SEM structure could be of real benefit to the reader.

·         2.3:  Most importantly, I think it’s important to describe the standards for drawing conclusions.  How do you determine what variables have effects on qol?  How do you determine what variables are mediators of the relationship between social media and qol?  Just saying that you’re using "SEM" isn't enough here.

·         In terms of patient experience, it seems to me that there is a big difference between, e.g., Stage 0 and Stage 3 cancers, especially some three years after surgery, when low-stage patients might be essentially cured, but higher stage patients might have recurred or still have a high chance of recurrence.  It seems as if the authors ought to pay more attention to these differences, and perhaps even adjust for them in analysis.

·         The main Results (Table 3 and associated text) should be presented in a way that gives a better idea of what each of the coefficients actually represents.  Again, a diagram could be really helpful.  It’s also not totally clear how many separate analyses these various coefficients are derived from.

·         I take it that the Bootstrapping SE is a standard error of the indirect effect?  And then the p-value is significance of the indirect effect?  That could certainly be made clearer.

·         I’m curious whether the authors can provide better justification for why their results represent mediation, rather than confounding (soc media is associated with anxiety, e.g., but doesn’t cause it) or even reverse causation (people with anxiety are drawn to social media).  A safer interpretation would just be that there is no direct effect of soc media on qol, after adjusting for the effects of the other variables.

·         The first par of the Discussion is pretty uninspiring.  Try to convey what it all means.

·         I would include variable timing of data collection (relative to surgery, covid, etc) and variable severity of disease among the limitations.

·         The conclusions drawn by the article seem a bit surprising and unsupported by the work.  If soc media leads to reduced qol via MH and sleep variables, then perhaps the real idea would be to monitor the use of soc media or at least to encourage use of social media in a way that does not cause anxiety, depression, or sleep issues.  It also just seems to me that the effects of the latter variables must be well known, and that if soc media isn’t included in the picture, then I’m less sure of what is novel here.

Author Response

Itemised reply to Reviewer 2

This article provides strong evidence for relationships between social media use, mental health and sleep measures, and quality of life in a population of breast cancer patients under post-surgery surveillance.  The topic is an important one, and it addresses important patient factors for recovery in a significant patient population.  The sample size is sufficient, the measures are appropriate, and the analysis appears to have been carried out correctly.

I do think that the authors could make the article much more approachable and impactful, by better describing the context and background and by better displaying and describing the results.  As written, the article feels like a pretty run-of-the-mill analysis of typical survey measures; but considering the unique population studied, as well as the novelty of studying social media use, I don’t think that this needs to be the case.  Please see below for some more detailed suggestions.

  1. Lines 46-51.  The end of this first par is a bit convoluted.  Better to start with a simple statement (and refs) that mental health problems are more common in cancer.  (As in the next par.)  Then move on to why that might be the case and what the consequences are.

Reply: We thank the reviewer for the candid assessment and comment. We have revised the last sentence by simplifying the sentences as suggested.

Lines 43-50: “The high fatality rate combined with treatment complications and socio-economic burden may leave the patient with wide-ranging psychosocial challenges [1-3]. Hence, mental health problems are common in cancer [4,5]. For instance, the cancer diagnosis itself may leave the patient distressed, especially when the diagnosis is poorly presented to the patient and without proper counselling [2,4,5]. This may substantially affect their mental health, quality of life and wellbeing [2,4] even for those patients without a history of mental illness [6].”

  1. 52.  “most patients with cancer are at risk”   Are all patients at risk?  Do most patients develop mental illness?  I’m not sure I see why most, but not all, would be at risk.

Reply: We thank the reviewer for the comment. It is supposed to be “all” instead of “most”. We have revised the sentence to reflect that.

Lines 53-54: “Evidence is well established that all patients with cancer are at risk of developing mental illness [4,6,7].”

  1. Lines 63-65.  The claim that holistic approaches should be adopted is not really supported.  I think you could support that systems should monitor MH and social media use; but it’s not clear that that entails ‘holistic.’

Reply: We have revised the sentence based on the reviewer’s recommendation. Thank you.

Lines 64-68: “Therefore, clinicians should adopt a system that monitors the mental health conditions of patients so as to benefit from prompt intervention which will further improve their well-being and quality of life (QoL).”

  1. Lines 66-76.  I think it would help to make a clearer distinction between the concepts of MH and QoL, and to explain the relationship, so that this doesn’t feel like repetition.  Certainly some QoL items are, basically, about MH. 

Reply: We thank the reviewer for the comment. We have revised the sentence by clarifying what QoL reflects in this study.

Lines 72-74: “Besides, mental illness has its fair negative effect on QoL of patients [23,24] as QoL (as used in this study) reflects the effect of an illness on an individual’s everyday life.”

  1. Lines 86-88.  If introducing covid, then it seems important to more directly make a case for either increased use of social media by patients during the pandemic and/or increased impacts of use.  The introduction of covid here feels a bit unfocussed.

Reply: We thank the reviewer for the comment. We have revised the COVID-19 part of the introduction by directly making a case for the increased use of social media.

Lines 81-83: “The current coronavirus 2019 (COVID-19) pandemic situation is aggravating the already dire healthcare needs of patients with cancer [12] due to the increased use of social media by patients as a medium of communication and/or for information [28-31].”

  1. Line 89-95.  Seems like you should provide some references for effects of social media on these various outcomes.  I’m sure they’re out there; but you haven’t particularly covered that angle.

Reply: We thank the reviewer for the comment. The reviewer is correct in pointing out that there are associations between the various variables used in the study. However, among cancer patients, there is no known study examining the “mediating role of anxiety, depression, sleepiness, insomnia, and sleep quality in the association between problematic social media use and QoL”. Please see the reported associations between the variables including problematic social media use and the other outcomes.

Lines 60-62: “Among patients with cancer, there have been significant associations between mental health variables such as anxiety, depression, sleepiness, insomnia, and sleep quality [12,18-20].”

Lines 74-76: “Furthermore, there have been significant associations between QoL and mental illnesses such as anxiety [12,16,20,25], depression [12,16,20,25,26], sleepiness, insomnia [16,26], and sleep quality [20,27].”

Lines 90-92: “It is also known that problematic social media use is associated with mental illnesses [37,39-43] and QoL [27].”

  1. It’ not clear how the timing of data collection and disease even overlaps with the covid-19 pandemic.  It appears that many of the participants would have been in the hospital before covid.  And data collection is long after hospitalization, when some patients are probably recovering nicely.  If covid really is a factor, then I think the authors should try to address that timing issue, as well as the heterogeneity of their sample in that respect.

Reply: We thank the reviewer for pointing out this important detail. Actually, the data collection period was the last sentence stated in the “design, participants, and procedure” section of the “materials and methods”. We have changed its position to the last but one sentence of the “design, participants, and procedure” section of the “materials and methods”. Please see the sentence below.

Lines 108-113: “All patients signed the informed consent before data collection. The data collection period was between March 2020 and November 2021. During this period, COVID-19 prevention policies such as restricted physical contact were in place and strictly adhered to. This study was approved by the ethics committee of Qazvin university of medical sciences (IR.QUMS.REC.1398.082).”

  1. 3:  It’s advisable to at least consider variable distributions, and to think about whether some non-parametric tests or transformations might be called for.

Reply: We have added the results of Mardia’s multivariate normality test to the data analysis section.

Lines 163-166: “The multivariate normality test was performed using the Mardia test [62]. The results of the multivariate normality test were not statistically significant for both skewness and kurtosis (p>0.05), suggesting that the data were normally distributed.”

  1. 3:  A diagram depicting the SEM structure could be of real benefit to the reader.

Reply: We thank the reviewer for the suggestion. We have added a figure showing the association between problematic social media use, anxiety, depression, sleepiness, insomnia, sleep quality, and QoL (i.e., physical component summary and mental component summary).

Lines 220-224:

Figure 1. The mediation model via structural equation modeling (SEM) showing the mediating roles of anxiety, depression, sleepiness, insomnia, and sleep quality in the association between problematic social media use and quality of life among patients with cancer

  1. 3:  Most importantly, I think it’s important to describe the standards for drawing conclusions.  How do you determine what variables have effects on qol?  How do you determine what variables are mediators of the relationship between social media and qol?  Just saying that you’re using "SEM" isn't enough here.

Reply: We thank the reviewer for the comments. We first conducted a univariate analysis before running SEM analysis. Also, we used an intercorrelation analysis between QOL and other variables to find which variables could be included in the SEM model.

Lines 166-168: “A univariate analysis before running SEM analysis. Also, we used an intercorrelation analysis between QOL and other variables to find which variables could be included in the SEM model.

  1. In terms of patient experience, it seems to me that there is a big difference between, e.g., Stage 0 and Stage 3 cancers, especially some three years after surgery, when low-stage patients might be essentially cured, but higher stage patients might have recurred or still have a high chance of recurrence.  It seems as if the authors ought to pay more attention to these differences, and perhaps even adjust for them in analysis.

Reply: We thank the reviewer for the comment. We did adjust the results with all sociodemographic and clinical variables in Table 1. We have appropriately revised Table 3.

Line 208: “Note: Sociodemographic and clinical variables in Table 1 were adjusted for in the model”

  1. The main Results (Table 3 and associated text) should be presented in a way that gives a better idea of what each of the coefficients actually represents.  Again, a diagram could be really helpful.  It’s also not totally clear how many separate analyses these various coefficients are derived from.

Reply:  We thank the reviser for the suggestion. We have reported the figure to ease interpretation. Also, we have added footnotes to help differentiate data for direct and indirect effects.

Lines 208-210: “Note: Sociodemographic and clinical variables in Table 1 were adjusted for in the model

aBootstrapping SE and its p-value reflects the standard error and significance for only indirect effect”

  1. I take it that the Bootstrapping SE is a standard error of the indirect effect?  And then the p-value is significance of the indirect effect?  That could certainly be made clearer.

Reply: Yes, please. Bootstrapping SE means standard error of the indirect effect and the p-value is the significance of the indirect effect. We have added footnotes to Table 3 to help differentiate data for direct and indirect effects.

Lines 209-210: “aBootstrapping SE and its p-value reflect the standard error and significance for only indirect effect.

  1. I’m curious whether the authors can provide better justification for why their results represent mediation, rather than confounding (soc media is associated with anxiety, e.g., but doesn’t cause it) or even reverse causation (people with anxiety are drawn to social media).  A safer interpretation would just be that there is no direct effect of soc media on qol, after adjusting forthe effects of the other variables.

Reply: We thank the reviewer for the comment. Factoring in anxiety as a confounding variable is one possible way to examine the influence of problematic social media use on QoL. There have been reports of reverses and even bidirectional relationships which also emphasises the reviewer’s point (see reference 1). The present study wanted to examine the effect of problematic social media use on QoL via anxiety (and other mental health variables) based on previous studies that link them together (cited in the manuscript). This study is quite similar to a previous study that examined “mediating role of psychological distress (depression, anxiety, and stress) in the association between internet gaming disorder (IGD) and two health outcomes (insomnia and quality of life) among adolescents during this COVID-19 pandemic” (see reference 2).

Lines 90-92: “It is also known that problematic social media use is associated with mental illnesses [37,39-43] and QoL [27].”

Reference

  1. Morita, M., Ando, S., Kiyono, T. et al. Bidirectional relationship of problematic Internet use with hyperactivity/inattention and depressive symptoms in adolescents: a population-based cohort study. Eur Child Adolesc Psychiatry (2021). https://doi.org/10.1007/s00787-021-01808-4
  2. Fazeli, S., Zeidi, I. M., Lin, C. Y., Namdar, P., Griffiths, M. D., Ahorsu, D. K., & Pakpour, A. H. (2020). Depression, anxiety, and stress mediate the associations between internet gaming disorder, insomnia, and quality of life during the COVID-19 outbreak. Addictive Behaviors Reports, 12, 100307. https://doi.org/10.1016/j.abrep.2020.100307

  1. The first par of the Discussion is pretty uninspiring.  Try to convey what it all means.

Reply: We thank the reviewer for the suggestion. We have revised the first paragraph of the discussion to convey the main findings.

Lines 226-230: “The present study which examined the mediating roles of anxiety, depression, sleepiness, insomnia, and sleep quality in the association between problematic social media use and QoL among patients with cancer revealed that problematic social media use only indirectly affected QoL via anxiety, depression, sleepiness and insomnia but not sleep quality.”

  1. I would include variable timing of data collection (relative to surgery, covid, etc) and variable severity of disease among the limitations.

Reply: We thank the reviewer for the suggestion. We included the timing of data collection (relative to surgery, covid, etc) and variable severity of disease among the limitations.

Lines 276-278: “Furthermore, replication is recommended to enhance the understanding and generalisability of the findings. Similarly, factors that affected prompt and quality service delivery during COVID-19 and the number of surgeries were not reported in this study. This would have helped to situate the challenges patients with cancer faced during COVID-19.”

  1. The conclusions drawn by the article seem a bit surprising and unsupported by the work.  If soc media leads to reduced qol via MH and sleep variables, then perhaps the real idea would be to monitor the use of soc media or at least to encourage use of social media in a way that does not cause anxiety, depression, or sleep issues.  It also just seems to me that the effects of the latter variables must be well known, and that if soc media isn’t included in the picture, then I’m less sure of what is novel here.

Reply: We thank the reviewer for the comment. We have revised the conclusion section to reflect the results of the study as suggested.

Lines 288-292: “This implies that healthcare workers should pay attention to how patients depend on electronic devices, the internet, and social media in order to properly educate them on the potential effect of its addiction on their mental health and QoL.”